# Relevance of ω-6 GLA Added to ω-3 PUFAs Supplements for ADHD: A Narrative Review

**DOI:** 10.3390/nu14163273

**Published:** 2022-08-10

**Authors:** Jelle D’Helft, Riccardo Caccialanza, Emma Derbyshire, Michael Maes

**Affiliations:** 1Springfield Nutraceuticals, Managing Director, Research & Development, Veldenstraat 23, 2220 Heist-op-den-Berg, Belgium; 2Clinical Nutrition and Dietetics Unit, Fondazione IRCCS Policlinico San Matteo, 27100 Pavia, Italy; 3Nutritional Insight, Epsom KT17 2AA, UK; 4Department of Psychiatry, Faculty of Medicine, Chulalongkorn University, Bangkok 10330, Thailand

**Keywords:** attention deficit hyperactivity disorder (ADHD), brain function, brain structure, cognition, developmental, difference, gamma-linolenic acid (GLA), hyperactivity, neuro-immune, neuroinflammatory, ω-3 PUFAs, oxidative stress, wellbeing

## Abstract

The use of polyunsaturated fatty acids in Attention-Deficit/Hyperactivity Disorder (ADHD) and developmental disorders has been gaining interest with preparations containing different dosages and combinations. Gamma-linolenic acid (GLA) is an ω-6 fatty acid of emerging interest with potential roles as an adjuvant anti-inflammatory agent that could be used with ω-3 PUFAs in the treatment of ADHD and associated symptoms. A narrative review was undertaken to examine the potential role(s) of the ω-6 fatty acid GLA. PubMed, Google Scholar, and Scopus were searched to examine the potential role(s) of the ω-6 fatty acid GLA as (1) an antioxidant and anti-inflammatory agent, (2) a synergistic nutrient when combined with ω-3 PUFAs, and (3) a potential etiological factor in ADHD and its treatment. The results show that GLA exerts anti-inflammatory effects by increasing dihomo-gamma-linolenic acid in immune cells. ω-3 PUFAs, such as EPA and DHA, are often co-administered with GLA because these ω-3 PUFAs may prevent the accumulation of serum arachidonic acid in response to GLA administration without limiting the storage of DGLA in immune cells. The administration of ω-3 PUFAs alone might not be sufficient to effectively treat patients with ADHD and developmental disorders. Overall studies point towards a combination of EPA and DHA with GLA in a 9:3:1 ratio appearing to be associated with ADHD symptom improvement. A combination of PUFAs may lead to better outcomes.

## 1. Introduction

Polyunsaturated fatty acids (PUFAs) include two pathways of fatty acids—ω-3 and ω-6—which are both known to play major biological roles as structural and functional components of cell membranes and have a profound influence on the development of the central nervous system [1]. As shown in Figure 1, linoleic acid (LA) and alpha-linolenic acid (ALA) are precursors of ω-6 and ω-3 families, respectively and are regarded as essential fatty acids (EFAs) as they cannot be synthesized by the human body in amounts needed for health and wellbeing, and thus need to be supplied by diet [2,3]. These parent fatty acids yield arachidonic acid (ARA, ω-6), eicosapentaenoic acid (EPA, ω-3), and docosahexaenoic acid (DHA, ω-3) which regulate body homeostasis and act locally via bioactive signaling lipids called eicosanoids [3].

Studies on evolutionary aspects of human diets indicate that major changes have taken place concerning types and profiles of dietary essential fatty acids obtained [4,5]. Genetically speaking, human beings in modern life live in a nutritional environment that differs from that which gave birth to their genetic pattern back in the evolutionary history of the genus *Homo* [5]. Whilst the dietary ratio of ω-6 to ω-3 fatty acids was once balanced and 1 to 1, in Western lifestyles this is now around 15–17 to 1 [4]. This is in stark contrast to recommendations from national agencies which advise that an ω-6 to ω-3 ratio of 4:1 is preferable [6]. It has been reported that lowering the ω-6 to ω-3 ratio to such thresholds could reduce enzymatic metabolic competition and facilitate the metabolism of more downstream products from ALA [7].

According to the Diagnostic and Statistical Manual of Mental Disorders V Edition (DSM-V), neurodevelopmental disorders (NDDs) are defined as a group of conditions with onset in the developmental period, inducing deficits that produce impairments of functioning [8,9]. NDDs comprise intellectual disability (ID), communication disorders, autism spectrum disorder (ASD), attention deficit hyperactivity disorder (ADHD), neurodevelopmental motor disorders, and specific learning disorders [8].

According to the DSM-V, ADHD diagnosis is based on some age-dependent symptoms of inattention and/or hyperactivity/impulsivity that interfere with functioning of development and should occur for at least 6 months [9]. ADHD is a neurodevelopmental disorder with onset in childhood, however, 50% of subjects continue to experience symptoms throughout adolescence and 30–60% in adulthood [10]. Overall pooled prevalence in the world is 7.2%; it is higher in males than in females, and it is most common in school-aged children [11]. Community prevalence of ADHD globally in children and young people has been reported to be between 2% and 7%, and an average of approximately 5% [12]. Other research reports a higher ADHD prevalence amongst boys, urban young people and those born to mothers with a history of psychiatric hospitalization [13].

These disorders typically manifest early in development, typically before a child begins school, and are characterized by developmental deficits that can impact social, personal, educational, or occupational functioning [9]. The developmental deficits can range from very specific limitations of learning or control of executive functions to global impairments of social skills or intelligence [1]. ADHD and ASD are amongst the most common neurodevelopmental disorders [1]. The impact of ADHD can have extended ramifications on academic performance, social skills, family relationships, and emotional wellbeing [14]. For some, without intervention, long-term outcomes can contribute to antisocial behavior, poor academic performance, non-medicinal drug use/addictive behavior, obesity, and reduced self-esteem [15].

The ω-3 PUFAs EPA and DHA have been considered for their potential roles in the treatment of symptoms in those affected by ADHD [16,17]. In the past, ω-6 PUFAs have been renowned for their generation of pro-inflammatory eicosanoids [3]. Subsequently, the potential benefits of ω-6 gamma linolenic acid (GLA) have been overshadowed. This standpoint changed in the 1980s when the potential anti-inflammatory effects of GLA began to generate more scientific interest [18,19,20]. It is now increasingly recognized that inflammation is a normal and innate part of human defense and tissue healing [21]. Renewed attention is therefore being given to ω-6 fatty acids, as they can regulate phagocytic capacity, cell migration and proliferation, and inflammatory mediator production [22]. This also includes a growing number of randomized controlled trials (RCTs) focusing on ω-6 fatty acids and aspects of cognitive and psychological wellbeing [23,24].

In this narrative review, we examine the potential role(s) of the ω-6 fatty acid GLA as (1) an antioxidant and anti-inflammatory agent, (2) a synergistic nutrient when combined with EPA, and (3) a potential etiological factor in ADHD and its treatment.

## 2. GLA as an Antioxidant and Anti-Inflammatory Molecule

GLA is composed of 18 carbon atoms with three double bonds and belongs to the category of ω-6 PUFAs [25]. GLA is present in human breast milk, a significant source for infants, and can be obtained from certain botanical seed oils and ingested from dietary supplements [25,26]. Natural sources of GLA include the oils of borage (*Borago officinalis* L., 20–26% GLA), black currant (*Ribes nigrum* L., 15–18%), and evening primrose (*Oenothera biennis* L., 8–12%) [25]. Some foods provide GLA in trace amounts, such as nuts and green leafy vegetables [26]. Unfortunately, GLA-rich foods are consumed in low quantities by the average person with the typical dietary intake of GLA being negligible [27].

Metabolically, as shown in Figure 1, GLA is produced in the body by the delta 6-desaturase enzyme acting on the parent fatty acid LA. It is then rapidly elongated to dihomo-gamma-linolenic acid (DGLA) by the enzyme elongase, acetylated, and incorporated into cell membrane phospholipids without resultant changes in ARA [28,29,30]. This has been demonstrated in in vivo metabolic research where GLA supplementation resulted in DGLA accumulation in neutrophil glycerolipids rather than ARA [31]. It has been postulated that this increase in DGLA relative to ARA within inflammatory cells such as neutrophils could diminish ARA metabolite biosynthesis [31]. This presents a plausible mechanism in terms of how dietary GLA could exert its anti-inflammatory effects [31].

Upon cell activation, DGLA is then released as free fatty acids (FAs) by phospholipase A2 and converted to several anti-inflammatory metabolites through competition with ARA for the enzymes cyclooxygenase (COX) and lipoxygenase (LOX) [32]. COX products from DGLA include prostaglandins 1 (PGE1), which can exert vasodilatory and anti-inflammatory actions [33,34]. 15-hydroxyeicosatrienoic acid (15-HETrE) is also the 15-lipoxygenase product of DGLA which can inhibit proinflammatory eicosanoid biosynthesis and exert anti-inflammatory properties [30,35]. The Δ-6-desaturase activity appears to be altered by various factors. For example, desaturase mRNA levels may be influenced by the quantity and composition of dietary carbohydrates, protein, fats, and micronutrients including vitamin A, B12, folate, iron, zinc, and polyphenols [36]. Reduced enzyme activity is also observed in elderly people [27]. Other work suggests that Δ-6-desaturase activity could also be reduced in young children presenting signs of dry skin, desquamation, and thickening of the skin, as well as growth failure [27].

GLA and its metabolites have also been found to influence the expression of several genes, regulating levels of gene products including matrix proteins [26]. Such gene products are thought to have central roles in immune function and programmed cell death (apoptosis) [26]. Critical associations are also proposed between ADHD and the level of oxidative stress which can induce cell membrane damage, changes in inner structure and function of proteins, and DNA structural damage which eventually culminate in ADHD development [37].

## 3. GLA Synergy with EPA

ARA can also be synthesized from DGLA through Δ-5 desaturase, encoded by fatty acid desaturase 1 (FADS1) within the FADS gene cluster (Figure 1). ARA and its potent eicosanoid products (prostaglandins, thromboxanes, leukotrienes, and lipoxins) play an important role in immune responses and inflammation [38]. Therefore, dietary supplementation with GLA has the capability to both increase levels of DGLA and its several anti-inflammatory metabolites, as well as ARA and its pro-inflammatory metabolic products and this might represent a therapeutic concern [25].

To counterbalance these pathways and make full use of the anti-inflammatory mechanisms of the molecule, the ω-3 long-chain-PUFAs (LC-PUFAs) EPA and DHA are often co-administered with GLA [39]. Humans supplemented with GLA, and EPA have substantially elevated blood EPA levels, but not ARA levels, suggesting that this supplement combination inhibits the development of pro-inflammatory ARA metabolites [39].

Evidence shows that 0.25 g/d EPA + DHA can block GLA-induced elevations in plasma ARA levels, while supplementation with borage + fish oil combinations inhibit leukotriene generation [40,41] and attenuate the expression of pro-inflammatory cytokines genes [42]. Furthermore, studies suggest that botanical oil combinations of borage oil, enriched in GLA, and echium oil (from *Echium plantagineum* L.) abundant in ω-3 PUFAs (ALA and stearidonic acids; SDA) enhanced the conversion of dietary GLA to DGLA whilst inhibiting the further conversion of DGLA to ARA [43]. Such supplementation strategies maintained the anti-inflammatory capacity of GLA, while increasing EPA, without causing accumulation of ARA.

This has been observed to directly translate into a clinical benefit in various therapeutic areas. For example, when enteral diets were enriched with marine oils containing EPA, DHA, and GLA, cytokine production and neutrophil recruitment in the lung were reduced, resulting in fewer days on ventilation and Intensive Care Unit stay-in patients with acute lung injury or acute respiratory distress syndrome (ARDS) [44]. Other research has observed decreased morbidity and mortality of critically ill patients with severe ARDS [45] and improved quality of life in asthma patients [40]. Positive outcomes of a combined ω-6/ω-3 PUFAs therapy have been observed in patients with rheumatoid and psoriatic arthritis in a RCT, where treatment with ω-3 LC-PUFAs and GLA led to an increase of GLA and DGLA concentrations in plasma lipids, cholesteryl esters, and erythrocyte membranes, indicating a reduction in the production of ARA inflammatory eicosanoids [46].

## 4. GLA and ADHD Focus

Even though the pathogenetic pathways of NDDs are still not completely clear, growing scientific evidence points to oxidative stress and neuroinflammation as triggers for their genesis, and might be pivotal in their clinical pattern and evolution [47,48]. ω-6 and ω-3 PUFAs and their metabolites are involved in immune-inflammatory and brain structural mechanisms and therefore these molecules may play a role in neurological disorders in which disruption of these mechanisms represent a contributing pathogenetic factor [49,50].

### 4.1. Pathogenesis and Role of Neuroinflammation

The etiology of ADHD is multifactorial and still not fully understood. Evidence indicates that ADHD is heritable, however, no genes of major effect have been detected so far [51]. Pre- and peri-natal inflammatory factors that can affect the in-utero environment seem to be also involved in the etiology of the disease i.e., infections [52], smoking [53], obesity and poor diet [54], and pollutants to which the mother might be exposed [55,56]. Prenatal exposures to inflammation have also been associated with a volume reduction of cortical areas associated with ADHD [57]. Furthermore, a bilateral decrease in grey matter volume in the cingulate and parietal areas was observed in children with ADHD [57].

The hypothesis that inflammation is part of the pathway to ADHD and more in general to NDDs is consistent. Neuroinflammation influences brain development and subsequent risk of neurodevelopmental disorders through mechanisms such as glial activation [58], increased oxidative stress [59], aberrant neuronal development [60], reduced neurotrophic support [61], and altered neurotransmitter function [62]. Studies have identified associations between ADHD and regulatory genes involved in cell adhesion and inflammation, such as the gene for the interleukin-1 receptor antagonist (IL-1 RA) [63]. Moreover, immune disorders such as eczema [64], asthma, rheumatoid arthritis, type 1 diabetes, and hypothyroidism [65] are associated with greater rates of ADHD diagnosis.

Higher levels of antibodies against basal ganglia [66] and dopamine transporter [67] have been detected in subjects with ADHD. Furthermore, ADHD patients have reported increased cerebrospinal fluid levels of pro-inflammatory cytokine tumor necrosis factor β (TNF- β) and lower amounts of anti-inflammatory cytokine IL-4 [68]. This translates into an overall vulnerability of central nervous system (CNS) structures. Under normal conditions the blood–brain barrier (BBB) separates the peripheral immune system, preventing peripheral immune cells from entering the CNS. However, injuries, inflammatory diseases, and psychological stress may compromise the BBB, consequently, peripheral activated monocytes and T-lymphocytes may penetrate the brain, thereby inducing microglial activation, neuroinflammation, and neurotoxic responses [69].

A lack of dopamine in ADHD pathogenesis has been supported by the evidence that medications like methylphenidate and amphetamines, which increase the levels of dopamine in the synaptic cleft, can temporarily improve symptoms [70]. However, up to 30% of ADHD patients do not respond to this treatment, while only about 50% show signs of improvement [71]. Recent studies have provided evidence for the role of the monoamine serotonin (5-HT), synthesized from the essential amino acid tryptophan in a few areas of the brain such as the dorsal raphe nucleus, and which is an important regulator of behavioral inhibition [72]. Children with ADHD may have lower blood levels of 5-HT [73], which might contribute to the symptoms of ADHD [74]. Therefore, alternative pharmacological treatments for ADHD include selective-serotonin reuptake inhibitors (SSRIs), serotonin-norepinephrine reuptake inhibitors (SNRIs), and tri-cyclic-antidepressants (TCA) all of which target the 5-HT system [75].

Relevantly, some ω-3 PUFAs and some ω-6 PUFAs such as GLA are directly involved in the synthesis, release, and re-uptake of neurotransmitters [76], Subsequently, an ω-3 PUFAs deficiency or an imbalance between ω-6 and ω-3 PUFAs may lead to impaired neurological functioning and behavioral disturbances similar to those which characterize ADHD [76].

### 4.2. Combined ω-6 and ω-3 PUFA Supplementation

As previously described, ADHD is a multifactorial condition, which depends on genetic and environmental factors [77,78]. Dietary deficiencies and imbalances have been attributed to the etiology of ADHD and are viewed as a plausible adjunctive therapy [79,80]. In particular, ω-3 PUFAs are well-recognized nutrients for being central to proper brain development and function [81].

The human brain is comprised of around 60% of lipids and essential PUFAs play a key role in its structure and functions [82]. Postmortem measurements of the cortex have shown that DHA concentrations in a healthy brain accrues and increase until 18 years-old [83]. Subsequently, ω-3 PUFAs shortfalls in the early life course are thought to have lasting impacts on the central nervous system, including reduced neuron size [83]. Neuron membranes need high amounts of ω-3 and ω-6 PUFAs, especially DHA and ARA, whose presence affects their fluidity, neurotransmission, and permeability, as well as membrane-bound proteins [81].

Over the last decade, several systematic reviews and meta-analysis publications have focused on the roles of ω-6 and ω-3 PUFA supplementation as adjunctive therapy for ADHD [84,85,86]. Derbyshire et al. (2017) undertook a systematic review of 16 RCTs finding that four studies [87,88,89,90] used supplements containing a 9:3:1 ratio of EPA/DHA/GLA which improved erythrocyte levels and showed promise as adjunctive therapy to traditional medications, lowering the dose and improving compliance with medications such as methylphenidate (MPH) [84]. Puri et al. (2014) conducted a meta-regression analysis of RCTs finding that longer study duration, GLA, and the interaction between GLA and EPA were statistically significantly linked to reductions in inattention in children with ADHD [86]. Gillies et al. (2012) reviewed 13 trials focusing on ω-3/ω-6 supplementation in children and teenagers with ADHD concluding that some data did show an improvement in attention and behavior symptoms [85]. It was, however, concluded that small sample sizes, variability of selection criteria, follow-up times, and the form and dosage of the supplement could have impacted on results [85].

A growing number of trials have also been published. A PubMed search limited to the last 10 years and focusing on studies in childhood and adulthood yielded the following main publications. Döpfner et al. (2021) found that a 4-month intervention with ω-6/ω-3 fatty acids in preschool children at risk of ADHD had some positive effects on symptoms although larger studies are warranted [91]. Chang et al. (2019) undertook a longer 12-week double-blind RCT, finding that EPA supplementation taken by 6–18-year-olds significantly improved blood erythrocyte levels and emotional symptoms, particularly amongst those with low EPA levels at baseline [92]. Interestingly, Barragán et al. (2017) found that children with ADHD treated with methylphenidate (MPH) required lower doses of the prescription medicine when given ω-3/ω-6 PUFAs supplements in a dosage of 558 mg EPA, 174 mg DHA, and 60 mg GLA (9:3:1 ratio) for 12 months, and experienced fewer medication-related side effects, implying that ω-3/ω-6 PUFAs may act as a useful adjunctive therapy to MPH, helping to improve tolerability, dosing, and adherence [87]. A placebo-controlled RCT recruiting 76 males (aged 12–16 years) with ADHD found that 12-weeks of supplementation with an ω-3/ω-6 PUFA supplement improved EPA, DHA, and total ω-3 fatty acid levels although no distinct benefits were found for psychological outcomes [88]. Johnson et al. (2012) randomized 75 children (aged 8 to 18 years) to 3 months of ω-3/ω-6 (Equazen eye q) or a placebo and found that responders (those with more than a 25% reduction in ADHD symptoms) had a significantly greater increase in ω-3 plasma levels and a reduced ω-6:ω-3 ratio at 3 and 6-months follow-up [89].

## 5. Discussion

Taken together ADHD is a prevalent condition, often persisting into adulthood that if left without intervention could become a risk factor for educational underachievement, unemployment, mental health disorders, and criminality [12]. Certain dietary factors but particularly supplementation with PUFAs appear to have mediatory effects on behavior in children with ADHD [93]. Unfortunately, habitual intakes of these fatty acids appear to be lacking from dietary sources. For example, it has been reported that in the United Kingdom only around 7.3% of children, 12.8% of teenagers, and 15.6% of young adults meet oily fish recommendation, one of the main dietary sources of ω-3 fatty acids [94]. As mentioned, GLA-rich foods are also only consumed in low quantities with the typical dietary intake of GLA being negligible [24]. It is, however, recognized that ongoing dietary intake studies are needed as it is difficult to accurately assess intakes of fatty acids and habitual intakes can be susceptible to under-reporting and plasma fatty acids are not always regarded as being accurate markers of food intake [95]. From a metabolic perspective, deficiencies of LA, ALA, ARA, EPA, and DHA can indicate malnutrition and deficiency of certain minerals, trace elements and vitamins which are important co-factors for the appropriate activity of desaturases [96]. Consequently, it is reasonable to assume that GLA, DGLA, AA, EPA, and DHA deficiencies can also be attributed to lower desaturases and elongase activities [96].

Given this, these fatty acids must be provided from alternative sources. This is particularly relevant to those with conditions such as ADHD who may have different physiological requirements for these PUFAs. For example, LaChance et al. (2016) found that children and young people with ADHD had elevated ratios of blood ω-6/ω-3 and ARA/EPA fatty acids suggesting an underlying metabolic disturbance in essential fatty acid levels in these individuals [97]. Common genetic variations within the fatty acid desaturase (FADS) gene cluster can also affect the rate of conversion of 18 C-PUFAs, including GLA, to LC-PUFAs [98]. This raises questions about whether gene-PUFA interactions may mean that “one size fits all” dietary recommendations and supplementation strategies may not be appropriate for all populations or indeed individuals within specific populations [25].

As discussed in the present publication, the pathogenetic mechanisms of NDDs including ADHD are not yet fully understood but clinical evidence points to oxidative stress and inflammation as potential triggering factors [37,65,72]. Since ω-3 and ω-6 PUFAs are involved in immune-inflammatory and brain structural mechanisms, these molecules may play a role in neurological disorders in which disruption of these pathways represents a contributing pathogenetic factor [29,49,82]. The anti-inflammatory functions of GLA, an ω-6 PUFA, are still to be fully understood but there is some evidence that ω-3 and ω-6 PUFAs may work better when combined. The most appropriate dosage ratio to express their therapeutic potential, however, is yet to be derived.

There is a growing body of studies indicating that specific combinations of PUFAs may be of benefit to ADHD management, potentially acting as an adjunctive to conventional medicines, possibly even lowering the dosage needed from these [81,84,87]. In particular, a growing number of publications demonstrate that the combination of EPA and DHA with the addition of GLA in a 9:3:1 ratio is associated with slightly better outcomes in terms of improvement in ADHD symptoms [87,88,89,90]. Subsequently, an emerging evidence base indicates that the presence of GLA in ADHD symptom treatment strategies might represent an added value. Furthermore, it is plausible that the administration of ω-3 PUFAs alone might not be sufficient to effectively treat patients with ADHD and developmental disorders and a combination of PUFAs may subsequently lead to better outcomes. Other work has found that ω-3 combined with ω-6 fatty acids, including GLA, may reduce ASD symptoms in children born preterm showing early ASD signs [99].

It is, however, recognized that ongoing research is needed. As mentioned, wide variability in sample sizes, selection criteria, intervention timeframes, and the form and dosage of supplement makes comparing results between studies challenging and could have skewed results [85]. The n-6 PUFA (GLA) evaluation techniques should be properly stated in research papers, and procedures should be uniform from clinical trials to facilitate cross-study comparisons.

In general, when interventions were shorter than 12-weeks duration and used lower supplement dosages findings are less highly regarded [100]. Erythrocytes only tend to survive in the body for 120 days, thus short supplementation trials may not be sufficient to detect changes in LC-PUFA compositions [88]. Furthermore, the turnover of PUFAs in the brain is thought to be slower in children meaning that longer periods of supplementation and/or higher doses are likely to be needed [100].

Finally, it is important to consider that there is growing interest in natural and sustainable vegan and vegetarian dietary sources of n-3 PUFAs [101]. For example, plant-based oils, marine fish oils and algal oils, which all contain various level of n-3 PUFAs may all have roles to play in improving ADHD symptoms. Validated methodologies are needed to develop and build research in this area.

## 6. Conclusions

In summary, PUFAs have been gaining attention in terms of their role in the etiology, treatment, and management of NDDs including ADHD. The evidence for ω-3 PUFAs is well documented but the role(s) of GLA are less well known. The present narrative review has explained that GLA appears to have anti-inflammatory properties that could be used with ω-3 PUFAs in the treatment of ADHD and associated symptoms. In particular, a combination of EPA and DHA with the addition of GLA in a 9:3:1 ratio appears to be associated with improvements in ADHD symptoms. It is possible that ω-3 long-chain PUFAs may prevent the accumulation of serum ARA in response to GLA. While evidence broadly supports these findings, ongoing research is needed in the form of mechanistic studies and rigorous clinical trials.

## Figures and Tables

**Figure 1 nutrients-14-03273-f001:**
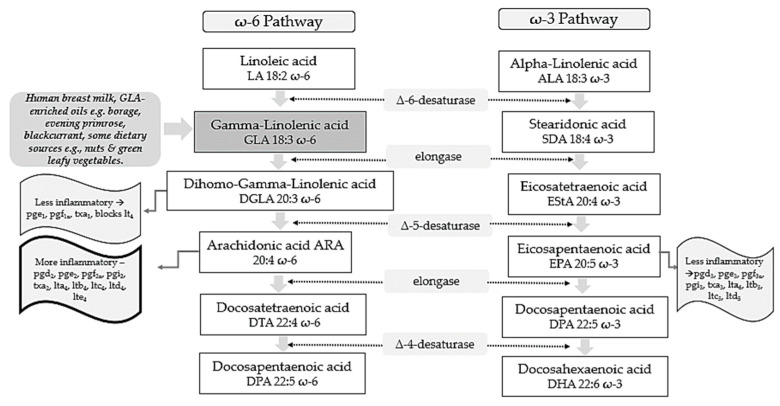
ω-6 and ω-3 family pathways. Key: pg, prostaglandin; pgi, prostacyclin; tx, thromboxane; lt, leukotriene.

## Data Availability

Not applicable.

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
