# Peer review of "Relevance of ω-6 GLA Added to ω-3 PUFAs Supplements for ADHD: A Narrative Review"

_nutrients, 2022, doi:10.3390/nu14163273_

Round 1
Reviewer 1 Report
Authors wrote an interesting review regarding the relevance og GLA supplements on ADHD.
Regarding the clinical aspect of ADHD it is clear, but there are some concerns regarding fatty acids.
1) pg2, line 96 "GLA is an EFA", is not exactly, GLA is synthetized from our organism, so it is not an EFA like LA or ALA.
2) pg3 line 118 "COX....thromboxane A1" are you sure that TXA1 is a metabolite of DGLA and not only of arachidonic acid? Please provide a reference
3) pg3 reference 33. It is a review, please change it with the specific article.
4) pg3 line 119 "LOX product of DGLA is... prostaglandin E1", but it is in contrast with line 117 "COX product of DGLA include prostaglandin 1", so PGE1 is the product of COX and LOX?
5) pg 3 line 121. The sentence (The D-6 desaturase...) is unrelated to previous one, please write a sentence to link the part of DGLA metabolism and the omega-6 pathway
6) pg 4, line 151. "neutrophils, lack d5 desaturase [37]". Is correct that reference 37 reports that neutrophils lack of D5, but it is not the focus and/or results of article, it reports that some studies shows that... but without any reference regarding it. Please report a correct reference or change the sentence.
Author Response
Thank you, our responses are attached in the following word document.

Reviewer 2 Report
Neurodegenerative diseases constitute a major problem of public health that is associated with increased risk of mortality and poor quality of life, especially in early life neuro defect population i.e. ADHD. Malnutrition is considered as a major problem that worsens the prognosis of patients suffering from ADHD and other neurodegenerative diseases. In this aspect, the present review is aimed to critically collect and summarize all the available clinical data as far as concern the clinical impact of n-6 & n-3 PUFA supplementation in neuropsychiatric diseases, highlighting on the crucial role of PUFAs nutritional supplementation in ADHD disease progression and management. According to the currently available animal and clinical data, lots of marine nutritional components seem to play a very important role in the protective capability on the progression of sleep related neuropsychiatric diseases.
In this review article, authors expression the updated nutraceuticals from PUFAs. This is a well-reviewed article on potential n-6 & n-3 PUFA in neuropsychiatric diseases, especially in ADHD. However, some considerations raised. The mentioned n-6 PUFA (GLA) assessment methods in reviewed papers should be consistent for clinic study. For a perspective aspect for disease prevention, some update potential biochemical pathway and markers should be discussed, especially the interaction among GLA, ALA, EPA and DHA. For example, plant base oil, marine fish oil and algal oil, which contain various level of n-3 PUFAs, the mechanism of anti-ADHD symptoms behaviors should be emphasized as well as the GLA in anti-ADHD actions. Overall, this is a well-organized review article.
Author Response
Thank you, our responses are enclosed in the attached word document.
